# Exploratory Systematic Review and Meta-Analysis of *Panax* Genus Plant Ingestion Evaluation in Exercise Endurance

**DOI:** 10.3390/nu14061185

**Published:** 2022-03-11

**Authors:** Shingo Ikeuchi, Mika Minamida, Touma Nakamura, Masatoshi Konishi, Hiroharu Kamioka

**Affiliations:** 1Research & Development Division, Maruzen Pharmaceuticals Co., Ltd., 1089-8 Sagata, Shinnichi-cho, Hiroshima 729-3102, Japan; m-minamida@maruzenpcy.co.jp (M.M.); t-nakamura@maruzenpcy.co.jp (T.N.); ma-konishi@maruzenpcy.co.jp (M.K.); 2Faculty of Regional Environment Science, Tokyo University of Agriculture, 1-1-1 Sakuragaoka, Setagaya-ku, Tokyo 156-8502, Japan; h1kamiok@nodai.ac.jp

**Keywords:** *Panax ginseng*, *Panax notoginseng*, meta-analysis, exercise endurance, ginsenoside Rg_1_

## Abstract

Background: Many studies that use food containing Panax genus plants (PGPs) have been conducted but most of them have not mentioned the effective compounds ginsenosides and their composition. Therefore, we conducted a systematic review and meta-analysis of time to exhaustion as an index of exercise endurance with ingestion of PGPs or ginsenosides to reveal their effects. Methods: We performed a systematic review with a comprehensive and structured literature search using seven literature databases, four clinical trial databases, and three general web search engines during 15–22 March 2021. A random-effects model was applied to calculate the standardized mean difference (SMD) and 95% confidence interval (CI) as the difference between the mean in the treatment and placebo groups. We evaluated the risk of bias of individual studies along with the risk of bias tool in the Cochrane handbook. This study was funded by Maruzen Pharmaceuticals Co., Ltd. (Hiroshima, Japan). The protocol for this study was registered with the UMIN-CTR (No. UMIN000043341). Results: Five studies met the inclusion criteria. The number of total participants was 90, with 59 in the ingestion-PGPs group and 64 in the control group, because three studies were crossover-design trials. We found that ingestion of PGPs or ginsenosides significantly improved exercise endurance (SMD [95% CI]: 0.58 [0.22–0.95], I^2^ = 0%). It was suggested that ginsenoside Rg_1_ (Rg_1_) and PGPs extract containing Rg_1_ were significantly effective in improving exercise endurance (SMD [95% CI]: 0.70 [0.14–1.27], I^2^ = 30%) by additional analysis. Conclusions: This systematic review suggests that the ingestion of PGPs or ginsenosides, especially Rg_1_, is effective in improving exercise endurance in healthy adults. However, further high-quality randomized controlled trials are required because imprecision and publication bias cannot be ignored in this systematic review.

## 1. Introduction

The *Panax* genus includes Korean ginseng (*P. ginseng* C. A. Mey.), Chinese ginseng (*P. notoginseng* [Burkill] F. H. Chen), and Japanese ginseng (*P. japonicus* C. A. Mey.) which are well known as ‘ginseng’ or herbal medicine plants traditionally. ‘Ginseng’ is named after its root shape, which is well branched and resembles the human body. While these plants have been used as important herbal medicines, especially in eastern Asia, they are also used in foods, drinks, dietary supplements, and other daily uses. Although ginseng usually needs more than four years to be harvested and used, a large amount of ginseng is grown and distributed. According to some statistics, there was approximately US $2085 million market for the worldwide sale of ginseng products in 2009 [1,2,3]. Other statistics and estimations reported that China, Korea, the United States, and Canada produced most of the fresh ginseng; over 80,000 tons of ginseng were distributed worldwide [3]. It is one of the well-known plant materials used in health products.

Ginsengs contain triterpene saponins, known as ginsenosides, which are particularly common in *Panax* genus plants (PGPs), and have been reported in about 40 species [4,5]. Over 30 ginsenosides are aglycones based on their carbon skeletons, the four-ring dammarane family, and are distinguished between protopanaxadiols and protopanaxatriols by the number of OH-moistures or sugars in their dammarane skeletons [6]. Because ginsenosides have some pharmacological effects and are diverse, ginsengs are called panacea or said to ‘regulate’ the human body and organs [4,5] even though some of the effects of ginsenosides are competitive. For example, ginsengs are reported to have functions which ameliorate vascular function and lipid profile, improve blood pressure, and have other effects [5]. In addition, it is traditionally said that ginseng relieves disorders such as fatigue, stress, and other physical or psychological dysfunctions. Although many studies have evaluated the pharmacological effects of ginsenosides, their actions in the human body are not clear because most of the studies were in vitro and in vivo animal trials.

Some studies have focused on new effects such as ergogenic aids. It is important in exercise to reduce the risks of obesity and related diseases such as metabolic syndrome and diseases of the cardiovascular, hepatic, and renal systems. If it is proved as effective in improving exercise performance, ginsengs would have advantage in its distribution and consumption with safety over other minor ‘ergogenic aids’ plants, such as Siberian ginseng, ephedrine, and ginkgo. Although there are several clinical trials, almost none of them mention its composition. A systematic review of *P. ginseng* [7] for exercise performance has already been reported, but it was not evaluated for exercise endurance and the components of each ginsenoside, which is an effective composition of PGPs, are not mentioned or considered. In particular, ginsenosides are rarely mentioned in studies that adopt commercially available ginseng products because most of them do not explain their contents other than the amount of total ginsenosides. Though the composition of ginsenosides is clear, more research on effects is needed.

We believe this study will provide new perceptions on the effects of PGPs and ginsenosides on exercise because there has been no systematic review on the clinical study of this topic. Therefore, we conducted a systematic review of time to exhaustion as an index of exercise endurance with ingestion of PGPs or ginsenosides to reveal whether PGPs can improve exercise endurance, and the effect of each component of ginsenosides.

## 2. Materials and Methods

### 2.1. Study Protocol and Registration

This systematic review was conducted in accordance with the Preferred Reporting Items for Systematic Reviews and Meta-Analyses (PRISMA) 2020 guidelines [8]. On 25 February 2021, the study protocol was confirmed by three researchers and a supervising coresearcher (H.K.) along with PRISMA protocol [9]. This protocol was registered in UMIN-CTR on 25 February 2021 (No. UMIN000043341, https://upload.umin.ac.jp/cgi-open-bin/ctr/ctr_view.cgi?recptno=R000049484, (accessed on 10 January 2022)). This systematic review was conducted with the research question “Does ingestion of plant extracts of Panax genus or ginsenoside improve exercise endurance of participants who have no problems (dysfunction or diseases) with motor and metabolic functions”, and the participant, intervention, comparison, and outcomes (PICOS) criteria. We made this article along with PRISMA 2020 Checklist (Appendix A). 

### 2.2. Eligibility Criteria

#### 2.2.1. Participant: P

The participants of this systematic review were adults (over 18 years old) who had no problems (dysfunction or diseases) with motor and metabolic functions, without children, and who were not pregnant (including the planning) or breastfeeding. Participants including persons who had dysfunction or diseases, and those with any acute or chronic diseases, such as diseases of the cardiovascular, pulmonary, endocrine, or gastrointestinal system, or other organ dysfunction or cancer, were excluded. Participants with diseases related to motor function of the limbs but diagnosed to have physical exercise ability by a doctor were included. 

#### 2.2.2. Intervention: I

The intervention was “acute or repeated periods of ingestion of foods containing plants, extract of *Panax* genus, or ginsenoside”. All types of food that are digested and absorbed by organs were included, but special forms, such as foods that avoid digestion and absorption, were excluded. We set the study duration from acute to 12 weeks because there are some studies which report that acute effects of ginsenoside are observed immediately after ingestion; a systematic review [7] reports a 12-week study as the longest.

#### 2.2.3. Comparison: C

The comparison was “acute or repeated periods of ingestion of foods which do not contain plants, extract of Panax genus, or ginsenoside”. If the study included no intervention as a control, we excluded it.

#### 2.2.4. Outcome: O

The primary outcome was endurance during exercise, and we evaluated the time of continuous exercise, which we call ‘time to exhaustion’. There were no secondary outcomes. We considered that PGPs do not affect cardiopulmonary function and output power because a systematic review [7] suggested that PGPs did not enhance physical performance, including VO_2max_ and output power. Thus, we thought it could have the effect of improving endurance by other mechanism(s) and we set time of continuous exercise as the primary outcome.

#### 2.2.5. Study Design: S

To collect studies widely, we conducted a systematic review of randomized controlled trial (RCT), quantitative RCT, and non-RCT.

### 2.3. Method of Literature Search

#### 2.3.1. Search Databases and Clinical Trial Registration Databases

We searched the literature along with the PICOS mentioned above, and we did not set limitations on year, language, and form by publication, even on retrieval of the complete article. However, we searched literature in English and Japanese because we could not understand other languages correctly. We also searched gray literature, including academic abstracts, administrative documents, and other forms of documents or literature.

The data on studies in English literature databases (Web of science, SCOPUS, CINAHL, PubMed, Cochrane Library, and AGRIS), English and Japanese literature databases (J-Dream III and Ichu-shi Web), clinical trial registration databases (ICTRP, ClinicalTrials.gov, ISRCTN, and UMIN-CTR), and general web search engines (Google, Yahoo! JAPAN, and Bing) were planned to be collected by reviewer S.I., searcher M.K., and a third-party searcher, and to be manually searched in journals possessed by reviewers S.I. and M.M. However, we conducted this search without a third-party searcher and searching for CINAHL to abide by the study protocol. We planned to contact the authors if we could, and it was necessary because, among other reasons, of the lack of data.

For completeness, we set search formulas and algorithms on each database with fundamental frameworks ‘(Panax OR ginseng OR ginsenoside) AND (athletic OR muscle OR exercise OR endurance OR ergogenic aid) AND clinical trial’. The details are provided in Table 1. Reviewers S.I. and M.M. manually searched independently on a journal possessed by Maruzen Pharmaceuticals Co. Ltd. We searched literature published during 15–22 March 2021 and last search date is provided in Table 1.

#### 2.3.2. Study Selection

We selected studies based on their title and abstract.

Exclusion criteria were:Reports on mixed results using other effective materials except PGPs or ginsenoside.The PICOS criteria is not met.The study is an in vitro or animal (except human) in vivo test.

After screening, each article’s content was checked to assess the selection criteria. Screening and checking articles were conducted by reviewers S.I. and M.M. individually, and selected if there were no conflicts. However, if a conflict of opinion occurred between the reviewers, the article was selected based on consensus with T.N. 

### 2.4. Data Collection and Meta-Analysis

#### 2.4.1. Data Collection

Data collection was conducted independently by reviewers S.I. and M.M. If there were any conflicts about the collected data, they were resolved by discussing with the researcher T.N. Reviewers S.I. and M.M. evaluated the quality of each study and the totality of evidence independently, and the researcher T.N. checked each result. Reviewer S.I. prepared the review manuscript based on the integrated results, and reviewer M.M. and researcher T.N. checked the contents. If there were any unknown or unclear outcome data, we attempted to contact the author.

#### 2.4.2. Data Items

We checked the following items in the literature: author, journal, title, PICOS, study setting, characteristics of participants, intervention, control, analysis, primary outcome, and key secondary outcome(s), with or without peer review. 

#### 2.4.3. Data Synthesis and Meta-Analysis

The primary outcome of this systematic review was endurance, which is indicated as exercise time (e.g., time to exhaustion). Therefore, we considered it to be a continuous variable and set the mean difference (standard deviation [SD]) of each group as an interval scale. Reviewers S.I. and M.M. conducted the meta-analysis using Review Manager 5.3 (RevMan 5) software (Cochrane, London, UK) for each plant species of intervention and outcomes. The meta-analysis was conducted using a random-effects model, standardized mean difference (SMD), and 95% confidence interval (CI). Heterogeneity was evaluated from the forest-plot with the value of I^2^ and the Cochrane Q value if it was suitable for application. 

#### 2.4.4. Risk of Bias of Individual Study

We evaluated the risk of bias of individual studies along with the risk of bias tool in the Cochrane handbook [10]. Reviewers S.I., M.M., and researcher T.N. were trained to evaluate the risk of bias by H.K., an expert of systematic review, and they checked their skill with each other to evaluate the risk of bias before conducting a systematic review, which produced satisfactory results. More specifically, we evaluated seven items: random sequence generation, allocation concealment, blinding of participants and personnel, blinding of outcome assessment, incomplete outcome data, selective reporting, and other biases. Reviewers S.I. and M.M. judged them individually as three grades; ‘low risk (0)’ was chosen if the publication resolved risks well because the study design was explained, ‘unknown risk (−1)’ was allocated if it retained risks because details were not mentioned appropriately, and ‘high risk (−2)’ was allocated if there were critical risks that could affect the study result through biased study design. After each evaluation, reviewers S.I. and M.M. checked each result and resolved discrepancies through discussion with researcher T.N.

#### 2.4.5. Risk of Bias of Across Studies

We fixed the risk of bias evaluation of each study and the total number of studies included in the systematic review and were evaluated independently by reviewers S.I. and M.M. Subsequently, indirectness was evaluated by reviewers S.I. and M.M. independently of whether the PICOS criteria were met. It was conducted as a risk of bias in systematic review along with the Grading of Recommendations Assessment, Development, and Evaluation (GRADE) system [11] that evaluated imprecision, inconsistency, and publication bias.

Imprecision was evaluated in all participants with the 95% CI. We did not have knowledge about the effect size of ginseng ingestion on endurance. Therefore, we calculated a suitable sample size with an effect size from the forest plot, α-error = 5%, and β-error = 10%. If the sample size from the included articles was less than a suitable sample size, we considered it as an occurrence of imprecision (−1). If the sample size was more than a suitable sample size but the 95% CI was wider than ±1/4 of the effect size, we also considered it as an imprecision (−1). 

Inconsistency was evaluated using the I^2^ value calculated in the forest plot and Cochrane’s Q value if it was suitable, along with the Cochrane handbook criteria. It was judged at three levels: ‘no heterogeneity (0)’ as I^2^ value of <30%, ‘unclear heterogeneity (−1)’ as I^2^ value of 30% to 60%, and ‘high heterogeneity (−2)’ as I^2^ value of >60%.

Publication bias was evaluated using a funnel plot, but if the number of included articles was not more than five, we considered it as unclear because of the lack of sufficient data, and we did not include it in the evaluation of total evidence.

Subsequently, the researcher T.N. checked the results of searching and evaluation by reviewers S.I. and M.M. and decided on the total evidence. 

#### 2.4.6. Additional Analysis

We planned to conduct additional analyses if the following conditions were applicable: (1) distinct diseases or healthy individuals; (2) distinct physically trained people (e.g., athletes, military) or non-trained people; (3) studies in which the intervention food was standardized or the amount of ginsenoside was analyzed; (4) studies without a large sample size; and (5) a distinct study duration (acute ingestion trial or continuous ingestion trial).

## 3. Results

### 3.1. Study Selection and Description of Included Studies

The results of the search are displayed in Figure 1. We collected 292 articles and clinical registrations (100 of them were duplicated), and we acquired 15 articles, of which we found that 10 articles did not meet PICOS criteria; therefore, we finally included five articles as listed in Table 2 [12,13,14,15,16]. The excluded articles are listed in the Appendix A. All trials measured exercise time to exhaustion, which is the time until participants were no longer able to maintain their exercise level. 

C.W. et al. [12] reported three randomized controlled crossover trials, but we collected data for related studies only. Twelve male (22.6 ± 0.6 years old) participants ingested food, including 5 mg of purified ginsenoside Rg_1_ (Rg_1_), which was obtained from *P. notoginseng*, at night before the exercise trial and an hour before exercise. The exercise was conducted using a cycle ergometer at 80% VO_2max_. The comparison was placebo food, which was an indistinctive intervention without Rg_1_, and the washout period was 4 weeks. In this study, Rg_1_ ingestion significantly extended exercise time to exhaustion (Rg_1_: 38.3 ± 6.7 min versus placebo: 31.8 ± 5.0 min).

Wu J. et al. [13] reported two randomized controlled crossover trials, but we collected data only on the related study. Twelve male (23.0 ± 0.5 years old) participants ingested food that included 5 mg of purified Rg_1_, from *P. notoginseng*, an hour prior to exercise. The exercise was conducted using a cycle ergometer at 80% VO_2max_. The comparison was placebo food, which was an indistinctive intervention without Rg_1_, and the washout period was 4 weeks. This study reported that cycling time to exhaustion was significantly increasing in Rg_1_ supplementation by 12% (Rg_1_:1364 ± 145 s versus placebo: 1219 ± 135 s).

Fadzel et al. [14] reported a randomized controlled crossover trial. Nine male (25.4 ± 6.9 years old) participants ingested food, including 200 mg of *P. ginseng* root extract, an hour prior to exercise. Exercise included running on a treadmill at 70% VO_2max_. The comparison was placebo food, and the washout period was 1 week. The duration of time to exhaustion was not significantly different between P. ginseng intake and placebo trials (*P. ginseng* ingestion: 88 ± 19.5 versus placebo: 84 ± 21.4 min).

Michael T.C.L. et al. [15] reported a randomized controlled parallel-group trial. Twenty-nine male and female (20–35 years old) participants ingested food with 1350 mg/day of *P. notoginseng* root extract before breakfast and dinner for 30 days. Exercise was conducted using a cycle ergometer at 70% VO_2max_ at the start, and a 30 W workload was added every 5 min. The comparison was placebo food, which is an indistinctive intervention without *P. notoginseng* extract. The ginsenoside components in the test foods were analyzed, and some of them were identified. In this study, *P. notoginseng* ingestion group significantly extended exercise time to exhaustion (*P. notoginseng* ingestion: 30.5 ± 12.8 min at baseline and 37.6 ± 10.2 min at after ingestion versus placebo group: 30.0 ± 12.6 min and 33.6 ± 10.5 min).

Allen J.D. et al. [16] reported a randomized controlled parallel-group trial. Twelve male and eight female (23.2 ±3.2 years old) participants ingested food including 200 mg/day of *P. ginseng* root extract 30 min prior to breakfast for 21 days. Exercise was conducted by a cycle ergometer with a 50 W workload being added every 2 min. The comparison was made with placebo food. Although the value of exercise time to exhaustion was not described, this study concluded that *P. ginseng* ingestion did not extend exercise time to exhaustion. We read the outcome figure and estimated values of the *P. ginseng* group as 11.4 ± 1.0 min at baseline and 11.9 ± 1.0 min after ingestion. We performed the same steps on the placebo group, and found they were 11.1 ± 0.5 min and 11.0 ± 0.6 min.

The included articles were considered to have low heterogeneity in the study design because they met the PICOS criteria of this systematic review. The participants were healthy adults and did not use or ingest medicines or dietary supplements. The intervention was ingestion of food, including purified ginsenoside or extract of *P. ginseng* or *P. notoginseng*. The control group ingested placebo food, and the outcome was exercise time to exhaustion. Four studies were conducted with a cycle ergometer and one trial used a running treadmill. Three trials had a fixed workload level at 70 or 80% VO_2max_ and two had increasing workloads. Thus, all trials were conducted with a heavy or severe load on the legs.

Three of the studies involved ingestion of food containing extracted or purified ginsenoside from *P. ginseng*, and two of the studies involved ingestion of food containing extracted or purified ginsenoside from *P. ginseng* or *P. notoginseng* as an intervention. Two of the studies involved acute ingestion of food 1 h prior to exercise containing extracts of ginseng or purified ginsenoside; one study was performed two times, once before night and once 1 h before exercise, and two studies were with continuous ingestion for 21 days or 30 days. 

### 3.2. Risk of Bias of Included Studies

The results of the risk of bias evaluation are shown in Figure 2. Each publication was described as an RCT, but most of them did not describe methods of randomization of participants, generating an allocation table, or blinding allocation table and food. We cannot disclaim selective reporting bias because none of the articles mentioned the study protocol or were registered in any clinical trial registration databases. While some risks of bias were found, we could not find any unfair conditions in all studies. Therefore, we concluded that the risk of bias of the included studies was ‘unknown, unclear risk (−1)’, and not high, but unclear about the impact of the study results. We evaluated that some situations are vague, but they are not unfair. We applied this result to the total risk of bias in the systematic review.

### 3.3. Data Synthesis Based on the Results of Literature Search and Meta-Analysis

We conducted the meta-analysis for SMD with random-effects model analysis as planned because we predicted that the included studies had a variety in food content and duration, as assumed. One article did not describe outcome data as values but as a graph, and we could not contact the author because contact information was not provided; therefore, we acquired outcome data values such as mean value and standardized division from the graph using computer programs (WebPlotDigitizer: https://automeris.io/WebPlotDigitizer/, (accessed on 10 January 2022)). 

These studies described the measured mean value and standard deviation, but not the amount and/or rate change from baseline; therefore, we calculated the change rate from the measured mean value and standardized division with baseline, which is measured before or after the study of each group, as 100% to apply to RevMan 5. 

We conducted a meta-analysis to evaluate each study outcome: endurance SMD [95% CI] was 0.58 [0.22–0.95], and the I^2^ value was 0% (Figure 3). This suggests that the ingestion of PGPs is significantly effective in improving endurance during ergometer exercise.

Meanwhile, our meta-analysis with distinguished origin plant (Figure 4a,b) showed SMD [95% CI] of 0.61 [0.12–1.09] and I^2^ value of 0% with *P. ginseng* ingestion, and 0.58 [−0.24–1.40] and 53% with *P. notoginseng* ingestion, respectively. The result of *P. ginseng* suggested a significant effect and was similar to the result of the total analysis, but that of *P. notoginseng* was not significant. However, the results of *P. notoginseng* displayed inconsistency because of the wide 95% CI which was caused by the low number of publications (only two were included). Therefore, it was difficult to conclude that *P. notoginseng* ingestion is not effective.

### 3.4. Additional Analyses

We conducted additional analysis on studies in which intervention food was standardized or analyzed the amount of ginsenoside, and with a distinct study duration (acute ingestion trial or continuous ingestion trial). Other additional analyses were not conducted because all studies included participants who were untrained for professional sports, military, or physically special occupations, and had no acute or chronic dysfunction or disease. We could not obtain detailed data and conduct each analysis. Additionally, all studies had similar sample sizes, and there were no studies with extremely large sample size.

#### 3.4.1. Additional Analysis for Ginsenoside Contents

We analyzed three studies; two studies included ingestion of purified Rg_1_, and one study included an extract of *P. notoginseng*, with the mentioned ginsenoside content ratio of approximately 1:1 of protopanaxatriol (which was described as sanchinoside R_1_, ginsenoside Rg_1_, and Re) to protopanaxadiol (which was described as ginsenoside Rb_1_ and Rd). The SMD [95% CI] was 0.70 [0.14–1.27], and the I^2^ value was 30% (Figure 5). This suggests that Rg_1_ improves exercise endurance, but it should be considered that there is medium heterogeneity, and only three studies were included.

#### 3.4.2. Additional Analyses for Study Duration

We analyzed three studies, and two studies were distinguished by duration; three studies were acute ingestion trials. Although C.W. et al. [11] reported ingestion twice, at trial day and before night, we considered that it is better that it included acute ingestion rather than continuous ingestion. The SMD [95% CI] was 0.77 [0.24–1.30], and the I^2^ value was 8% (Appendix A). While two studies included several weeks of continuous ingestion trials, the SMD [95% CI] was 0.38 [−0.15–0.91] and I^2^ value was 0% (Appendix A). 

This suggests that acute ingestion of PGPs is effective for improving exercise endurance, but not with continuous ingestion. However, their analysis was conducted with few studies.

### 3.5. Quality of Evidence

The indirectness of each article was evaluated as low bias (0), based on the PICOS criteria. All studies included healthy adult male/female participants, intervention foods included PGP extract or Rg_1_, control was ingestion of food that did not include PGP extract or Rg_1_, and outcome was measured as time to exhaustion with a cycle ergometer or running on a treadmill. Imprecision was evaluated as unclear bias (−1); although the total sample size was 123, which is over the calculated value by effect size (=0.58), α-error (=5%), and β-error (=10%), the 95% CI was 0.22–0.95, and was wider than 25% of the effect size. Inconsistency was evaluated as low risk (0), because the I^2^ value was 0%. It was not suitable to evaluate inconsistency with Cochran’s Q value by the reason of outcome, and time to exhaustion or exercise is a continuous value. We could not evaluate publication bias appropriately due to the low number of articles included.

In conclusion, evidence of this study was concluded to be of a ‘low’ evidence level, which is two levels down from ‘strong’, the top level according to the GRADE system, because of risk of bias of total studies and imprecision. However, publication bias was not yet clear.

## 4. Discussion

### 4.1. Ginseng, Ginsenosides, and Their Regulations

Ginsenosides are unique components of PGPs and are known for their pharmacological effects [4,5]; their amounts and ratios vary according to species, age, harvest location and season, region [17,18,19,20,21,22,23,24,25,26,27], and other processes in the production of *P. ginseng* products [28,29,30,31,32,33,34,35,36,37,38]. Using ungrown or incorrect materials may cause confusion in the homogeneity of quality; therefore, some products of PGPs were determined by the production method. For example, red ginseng, which is steamed and well-dried, is a form of *P. ginseng* used for various effects including relieving fatigue by improving enzyme activity of creatin kinase and anti-oxidation; a monograph reported that the raw material, root, and rhizome of *P. ginseng* should be ≥4 years old. Its production methods are registered with the International Organization for Standardization [39,40], and the manufacturing process includes cutting, cooking with vapor, drying, and detailed temperature and time conditions [39]. Red ginseng has been suggested and certified to display six functions of immunity improvement, fatigue relief, blood circulation improvement, aid in memory improvement, aid in antioxidant activity, and aid in menopausal women’s health, under the health food institution in Korea, but these mechanisms have not yet been established [39]. White ginseng, which is the dried form of *P. ginseng*, contains ginsenosides, but it differs from red ginseng in terms of species and ratio [41]. Despite this traditional and scientific knowledge, and PGPs being considered to have different types of functions with different forms, monographs of few official organizations mentioned total effects with no distinguished form of ginsengs [42,43]. Similarly, ginsenoside contents are rarely mentioned or described in clinical studies. For example, a systematic review by Hoang et al. [7] concluded that ingestion of *P. ginseng* or *P. notoginseng* has no effect on exercise, but the components of ginsenoside were not mentioned and considered in the review in almost all included studies. There are many dietary supplements or complementary health foods that are not standardized for each ginsenoside. This may be the result of by unwillingness of suppliers or goods of unstable quality; therefore, customers worldwide may suffer from such disadvantages. Therefore, a fair evaluation of the effects of PGPs and ginsenosides is important for its accurate use and safety.

### 4.2. Effects of PGP Ingestion

We included five studies that evaluated skeletal muscle endurance of the legs with healthy and untrained individuals for systematic review because each study measured the time to exhaustion in leg ergometer or treadmill exercise, which was estimated as heavy or severe extensive workload [44]. Researchers used foods that included extracts of *P. ginseng* or *P. notoginseng*, or purified Rg_1_ from these plants as an intervention, and it was shown extended the exercise time. Although some studies have conducted trials using extracts that analyzed the amount and ratio of each ginsenoside, there was a considerable number of trials using Rg_1_ and an extract of *P. notoginseng*, which contained the same amount of protopanaxatriol and protopanaxadiol ginsenosides, even though the inconsistency was unclear. 

### 4.3. Effects of Ginsenoside Rg_1_

Rg_1_ is one of the major components of the root or rhizome of *P. ginseng* and *P. notoginseng* [4,5]. For example, it is well known that it makes the sympathetic nerve predominant [39,45]; therefore, it may be involved in extending exercise time. It has also been reported to have anti-fatigue effects in mouse trials [46]. Rg_1_ has been reported to have anti-inflammatory effects [47] by suppressing the nod-like receptor family pyrin domain containing-1, interleukin (IL)-1β, and IL-18 [48]. It is considered to be important for improving exercise endurance because inflammation inhibits energy generation via metabolic carbohydrates and lipids in skeletal muscle. Rg_1_ has also been reported to improve the antioxidant effect on skeletal muscle [49] and liver [50], Alzheimer’s disease [51], neuroprotection [52], and anti-obesity [53], in in vitro and in vivo studies. 

Wu J. et al. [13] reported that ingestion of Rg_1_ suggested increased leucocyte infiltration into skeletal muscles during exercise; when decreased leucocytes infiltrate the skeletal muscle, cells indicate apoptosis markers, and suppress inflammatory collagenase expression after 3 h of exercise. The study reported that locally recruited neutrophils and IL-1β, which are expressed by neutrophils in skeletal muscle, are essential for enhancing muscle performance via exercise-induced translocation glucose transporter type 4 [54]. As mentioned above, Rg_1_ suppressed IL-1β, but it may suppress only overexpression, which induces inflammation, and not mild expression. Therefore, Rg_1_ was suggested to promote leucocyte infiltration into the skeletal muscle and improve muscle exercise endurance via glucose homeostasis during exercise and suppress inflammation after exercise. Hou et al. [12] also reported anti-inflammatory effects; it suppressed thiobarbituric acid after 3 h of exercise, tumor necrosis factor α (TNFα) messenger ribonucleic acid (mRNA) expression, and accelerated IL-10 mRNA expression immediately after exercise in skeletal muscle. Accordingly, ingestion of Rg_1_ has been suggested to have an anti-inflammatory effect and improve exercise endurance.

On the other hand, Rg_1_ is reported to make sympathetic nerves predominant and improve energy metabolism. Additionally, some in vitro and in vivo studies have suggested that Rg_1_ upregulates peroxisome proliferator-activated receptor γ [55,56], AMP-activated protein kinase [56,57] pathways, peroxisome proliferator-activated receptor gamma coactivator-1 (PGC-1), and mitochondrial uncoupling protein-1 [58], which are essential for energy metabolism during exercise [59], especially PGC-1, and were reported to be related to the anti-fatigue of skeletal muscles [60,61,62]. As a result, it is considered that Rg_1_ has anti-inflammatory effects and accelerates energy generation. Therefore, the effects of Rg_1_ on exercise endurance may be affected by multiple pathways. 

### 4.4. More Verification of Items

#### 4.4.1. Study Design

This suggests that acute ingestion or twice are effective on additional analysis, and it was considered that Rg_1_ could affect instantly but its effect may not accumulate. Wu et al. [12] concluded that Rg_1_ effectively eliminates senescent cells, but its effect may not be significant. It may be affected by ingestion times because Wu et al. [12], Hou et al. [13], and Fadzel et al. [14] conducted their study in which intervention food was ingested an hour prior to starting exercise, but other studies [15,16] reported ingestion before meals. Therefore, if more clinical trials are conducted, the timing of ingestion of test foods, including PGPs or ginsenosides, should be considered. 

In this study, especially additional analysis, we cannot verify the impact of ginsenoside(s), without Rg_1_, on the effect size of each study. In this study, high effect size was indicated by Wu J. et al. [12] and Hou C.W. et al. [13], in which purified Rg_1_ was used in the intervention. It may be considered as the effects of the other components with or against Rg_1_. Michael T.C.L. et al. [15] ingested an extract of *P. notoginseng* as an intervention, which contained Rg_1_ of approximately 50 mg/day, which is 1/4th of total ginsenoside. While it was equivalent to 10 times the amount considered in the Wu J. and Hou C.W. study interventions, the effect size was smaller. It was also indicated that certain workload trials had a larger effect size than continuously adding workload trials. It may be shorter than the true endurance because the workload increases with increasing exercise time.

#### 4.4.2. Ginsenosides Other Than Rg_1_

Some types of ginsenosides are reported to have an effect on accelerating energy metabolism. For example, ginsenoside Re, which is a panaxatriol ginsenoside, is suggested to be involved in energy metabolism because it has been reported to bind to the cardiac intermediate-conductance calcium-activated potassium (IKs) channel and its effects via binding to the sex receptor and activating endothelial nitric oxide synthase (eNOS) via the PI3 kinase/Akt-dependent pathway [63]. Although it occurs by binding to the sex hormone receptor (e.g., androgen, estrogen, and progesterone receptors), it does not activate genotropic action but activates the non-genotropic pathway which is described as the sex-hormone receptor-c-Src-PI3k-kinase-Akt-eNOS pathway [64] because ginsenoside Re-sex-hormone receptor cannot recruit a coactivator and it cannot activate the genotropic pathway [63]. It may be considered that other ginsenosides may affect in a similar manner or in a competitive pattern. As a result, we cannot verify the determinants of effect size because of the differences in study design, including trial duration, adding workload, ginsenosides, especially protopanaxatriols other than Rg_1_, and other factors including participants’ backgrounds and intervention foods. We hope more RCTs, which with clear contents of ginsenoside as the intervention, corresponding study design, and similar participants, will be conducted.

### 4.5. Limitations of This Study

This study has some limitations. First, we included only five studies, even though we performed a search exhaustively using literature databases, clinical trial databases, and general web search engines. There are an insufficient number of included articles, which causes imprecision and publication bias; therefore, evaluation evidence of this systematic review is at a ‘low’ level. Second, there are some risks of bias that we cannot disclaim; these studies were not registered to clinical trial registration databases or published proposal, and some studies were published more than 10 years ago. It is important to register the protocol of the clinical trials to clinical trial registration databases or published proposals for secure validation. It was also unclear whether the allocation table was randomized and concealed for participants and food. In this study, it was impossible to exclude selection bias because some studies were published before a measurement tool to assess systematic reviews was established [65]. Further, some studies did not describe methods of allocation, concealment, and other factors needed to reduce risk of bias. Therefore, the risk of bias remains unclear. Subsequently, the validity of the additional analysis cannot be proved appropriate; some analyses indicate considerable inconsistency. Finally, we cannot appropriately evaluate the components other than the Rg_1_ effect because some studies did not mention the components and amount of ginsenosides. These limitations warrant more clinical trials with a reliable study design, fair and detailed description, and large enough number of samples.

### 4.6. Safety Evaluation

In each piece of literature included in this study, PGPs, its extract, or ginsenoside were considered to be safe because no adverse events were reported. However, the participants were adults (20–40 years old), had no diseases, and consumed no medication. The safety of the ingestion of PGPs and ginsenosides for people who are elderly, have diseases, and take medicines should be considered because we cannot prove it. Improving exercise endurance is more important for older people than younger ones because exercise for a long time is essential to maintain their muscle, normal blood glucose and lipids, and other factors related to their quality of life. Therefore, we need to conduct more clinical trials, such as studies in elderly people, to certify its efficacy and safety.

## 5. Conclusions

We performed a systematic review to evaluate the effect of ingestion of PGPs or ginsenosides on endurance, and it is suggested to be effective. It is thought that Rg_1_ and other protopanaxatriols are active bodies for improving exercise endurance of skeletal muscle, but we could not verify the process that improves endurance and the effect of other ginsenosides. There are limitations that can be attributed to the small number of studies; further clinical studies with PGPs or ginsenosides for exercise should be conducted. 

## Figures and Tables

**Figure 1 nutrients-14-01185-f001:**
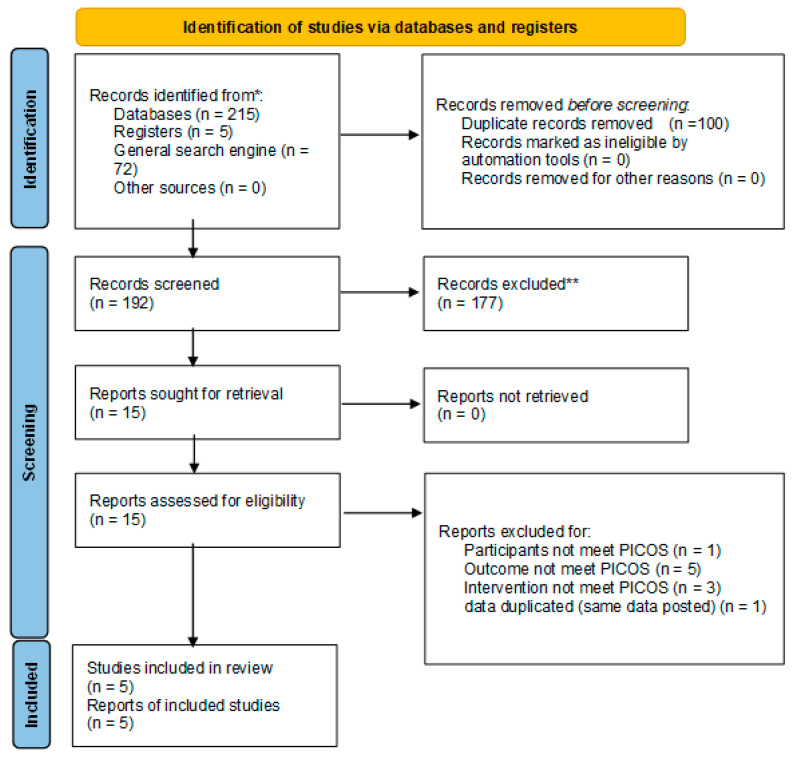
The results of the search studies.

**Figure 2 nutrients-14-01185-f002:**
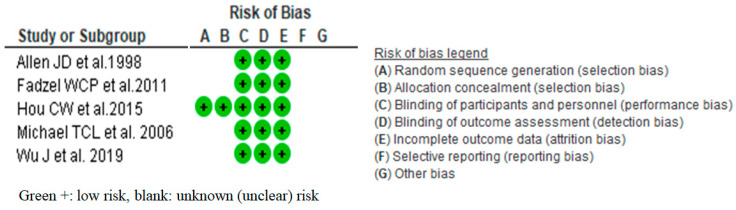
The result of evaluation risk of bias [12,13,14,15,16].

**Figure 3 nutrients-14-01185-f003:**
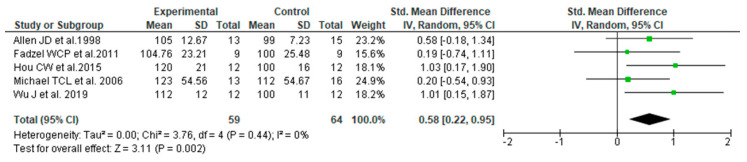
Forest plot for exercise endurance with ingestion Panax genus plants or ginsenoside [12,13,14,15,16].

**Figure 4 nutrients-14-01185-f004:**
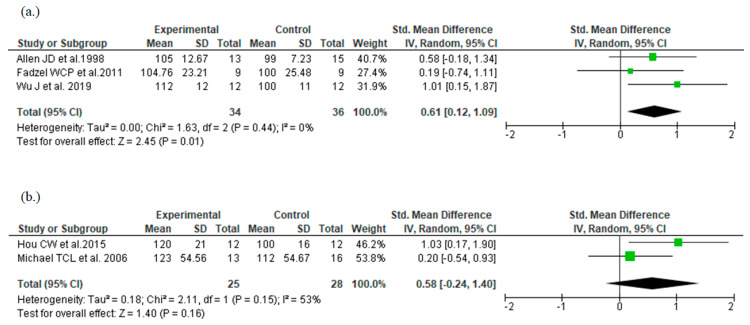
Forest plots for exercise endurance with ingestion each Panax genus plants species [12,13,14,15,16]. (**a**) ingestion *P. ginseng*; (**b**) ingestion *P. notoginseng*.

**Figure 5 nutrients-14-01185-f005:**
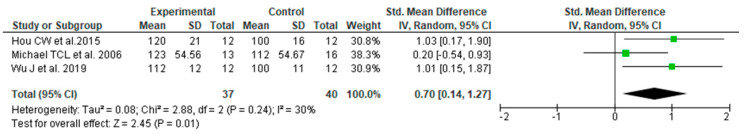
Forest plot for exercise endurance with ingestion Panax genus plants with ginsenoside compositions were standardized or analyzed [12,13,14,15,16].

**Table 1 nutrients-14-01185-t001:** Search formula for searching literature and search date.

Database	Search Formula	Last Search Date
PubMed	(Panax OR ginseng OR ginsenoside) AND (athletic OR muscle OR exercise OR ergogenic aid) AND clinical trial	15 March 2021
PubMed [Mesh]	(ginsenoside): ‘ginsenosides’. (athletic): ‘athletes’, ‘sports’ (muscle): ‘muscles’ (exercise):’exercise therapy’. ‘exercised’. ‘exerciser’. ‘exercisers’. ‘exercising’ (ergogenic): ‘ergogenicity’.’performance-enhancing substances’.’performance-enhancing’.’performance-enhancing substances’. ‘ergogenics’ (clinical trial): ‘clinical trials as topic’	15 March 2021
Cochrane library	(Panax OR ginseng OR ginsenoside) AND (athletic OR muscle OR exercise OR ergogenic aid)	15 March 2021
AGRIS	(Panax OR ginseng OR ginsenoside) AND (athletic OR muscle OR exercise OR ergogenic aid) AND clinical trial AND (English and Japanese)	17 March 2021
J-DreamIII (JMEDPlus, JSTPlus)	(Panax OR ginseng OR ginsenoside) AND (athletic OR muscle OR exercise OR ergogenic aid) # in Japanese	16 March 2021
Ichu-shi Web	(Panax OR ginseng OR ginsenoside) AND (athletic OR muscle OR exercise OR ergogenic aid) # in Japanese	15 March 2021
Web of science	(Panax OR ginseng OR ginsenoside) AND (athletic OR muscle OR exercise OR ergogenic aid)	15 March 2021
Scopus	(Panax OR ginseng OR ginsenoside) AND (athletic OR muscle OR exercise OR ergogenic aid)	15 March 2021
Clinical trial database	search with mixed keywords below; (Panax OR ginseng OR ginsenoside) AND (athletic OR muscle OR exercise OR ergogenic aid) # in English and Japanese	17 March 2021
General search engine	search with mixed keywords below; (Panax OR ginseng OR ginsenoside) AND (athletic OR muscle OR exercise OR ergogenic aid) # in English and Japanese	22 March 2021

#; searched in each Languages because some sources provided service with multi-Languages.

**Table 2 nutrients-14-01185-t002:** Studies included in systematic review.

Study ID	Study Design							
	Crossover/Pararell	Duration	Sample Size	Ingestion Food	Ginsenoside Composition	Exercise	Exercise Intensity	Other Measurement(s)
How C.W. et al., 2015 [12]	crossover	2 times: night before trial and before trial	12 male	5 mg of ginsenoside Rg_1_ (from *P. notoginseng*)	clear (95% purity of ginsenoside Rg_1_)	cycle ergometer	80% VO_2MAX_	citrate syntase activity, inflammatory markers
Wu J. et al., 2019 [13]	crossover	acute	12 male	5 mg of ginsenoside Rg_1_ (from *P. ginseng*)	clear (95% purity of ginsenoside Rg_1_)	cycle ergometer	80% VO_2MAX_	leukocyte infiltration in skeletal muscle, inflammatory markers
Fadzel W.C.P. et al., 2011 [14]	crossover	acute	9 male	200 mg of *P. ginseng* root extract	unknown (not mentioned)	run treadmill	70% VO_2MAX_	heart rate, VO_2_, core body/skin temperature, blood paramaters
Michael T.C.L.. et al., 2006 [15]	pararell	30 days	EXP: 13 CON: 16 male/female	1350 mg/day of *P. notoginseng* extract	partially clear (HPLC analysis of conducted, and some ginsenosides were quantified)	cycle ergometer	65–70% VO_2MAX_ at start, and added 30 W workload each 5 min	heart rate, VO_2peak_, skin blood flow
Allen J.D. et al., 1998 [16]	pararell	21 days	EXP: 13 CON: 15 male/female	200 mg/day of *P. ginseng* root extract	unknown (not mentioned)	cycle ergometer	Added 50 W workload each 2 min	heart rate, VO_2peak_

EXP: experimental group, CON: control group.

## Data Availability

The protocol of this study was registered at UMIN-CTR (No.:UMIN000043341, https://upload.umin.ac.jp/cgi-open-bin/ctr/ctr_view.cgi?recptno=R000049484, (accessed on 10 January 2022)) and have not been amended any information. More detailed information and the datasets generated during and/or analyzed during the current study are not publicly available due to limitations of the database and literature, but are available from the corresponding author upon reasonable request.

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
