# Peer review of "Exploratory Systematic Review and Meta-Analysis of Panax Genus Plant Ingestion Evaluation in Exercise Endurance"

_nutrients, 2022, doi:10.3390/nu14061185_

Round 1

Reviewer 1 Report

This is a systematic review on the use of ginseng and the possibility of improving endurance performance. Although few studies have been reviewed, the method is robust enough to indicate some finding. In conclusion, it is important to suggest experimental designs so that both ginseng and other compounds have more rigorous studies for future meta-analyses.
I suggest a more detailed review of the English language in terms of fluidity and academic form.

Reviewer 2 Report

Thank you for this interesting article. The search criteria and inclusion / exclusion were rigorous and well-defined. However, I have a few concerns, which I have outlined below. It is clear that the mechanistic action of ginseng is explained in great detail, but some more general information is missing. For example, the introduction does not list any purported functions of ginseng.

Secondly, time to exhaustion and exercise domain / intensity are not very well explained and need clarification.

Unless I missed it, at no point do you actually show the data from the studies outlining the time to exhaustion for the trials. This needs to be included to allow the reader a sense of suggested improvement.

I felt I couldn’t fully support your conclusion due to the lack of data shown from the studies. Unless this is included I would not support the publication as its conclusion seems overly positive considering the lack data shown.

Abstract:

Exercise endurance needs to be defined. Is this improved time to exhaustion, improvements in VO2max, increases in lactate threshold, improved running economy, reduced oxygen consumption?

Introduction:

Please include some content on the alleged benefits of ginseng ingestion (I’m aware some of this is shown in the discussion but the reader would benefit from some inclusion here).

Line 43: Your reference for ginseng sales in over 20 years old. Please find a more recent number.

Line 47: please re-phrase ‘some kinds of triterpene…’ do you mean they have some or different kinds of?

Line 61: what are ‘minor ergogenic aid plants’?

Line 72/73: ‘improve exercise endurance’ is very vague and needs to be explained / defined. What measures would these be?

Methods:

Line 108: so only time to exhaustion was used as a measure? The ability to exercise at a higher intensity for a given time should also be considered. Time to exhaustion is not necessarily the best measure of endurance ability as it is not ecologically valid. Exercising at the same intensity, but with lower O2 cost / heart rate / rating of perceived exertion would all be valid measures also. Faster time to completion for a set distance should also be considered. This seems quite limited.

The search and inclusion / exclusion criteria are described accurately.

Table 2:

How: ‘sintase’ should be synthase; ‘inframmatry’ should be inflammatory

Wu: ‘leukosyte’ should be leukocyte; ‘skeretal’ should be skeletal  

Fadzel: ‘tredmill’ should be treadmill    

Michael: ‘minuts’ should be minutes

Allen: ‘minuts’ should be minutes

Evaluation of the studies, risk of bias etc. are presented in good detail.

Discussion:

Line 378: please expand on the proposed functions of ginseng and how these might occur.

Line 403: not all studies used ergometers, there were treadmill runs as well. ‘middle or hard extensive’ is not a scientifically accurate way of describing exercise intensity. Work rates ay 70-80% VO2max are in the ‘heavy / severe’ exercise intensity domain for untrained subjects and will be above the lactate threshold for most. In some studies the intensity was increased until volitional exhaustion and as such was ‘maximal intensity exercise’.

Line 415: trials not ‘trial’

Line 416: effects not ‘effect’

Line 443: ‘anti-fatigueness’ is not a correct term

Line 449: I do not understand what you are trying to say in this sentence. Please re-phrase.

Round 2

Reviewer 2 Report

thank you for addressing the suggestions. there are still minor spelling errors and I feel that the endurance performance / exercise intensity domains could be explained better. 

however, these minor issues do not detract from the main outcomes and i'm happy to support the publication.